# BackdoorBench: A Comprehensive Benchmark of Backdoor Learning

**Baoyuan Wu**[1]*  **Hongrui Chen**[1]  **Mingda Zhang**[1]  **Zihao Zhu**[1]
**Shaokui Wei**[1]  **Danni Yuan**[1]  **Chao Shen**[2]

[1]School of Data Science, Shenzhen Research Institute of Big Data,
The Chinese University of Hong Kong, Shenzhen
[2]School of Cyber Science and Engineering, Xi'an Jiaotong University

## Abstract

Backdoor learning is an emerging and vital topic for studying deep neural networks' vulnerability (DNNs). Many pioneering backdoor attack and defense methods are being proposed, successively or concurrently, in the status of a rapid arms race. However, we find that the evaluations of new methods are often unthorough to verify their claims and accurate performance, mainly due to the rapid development, diverse settings, and the difficulties of implementation and reproducibility. Without thorough evaluations and comparisons, it is not easy to track the current progress and design the future development roadmap of the literature. To alleviate this dilemma, we build a comprehensive benchmark of backdoor learning called *BackdoorBench*. It consists of an extensible modular-based codebase (currently including implementations of 8 state-of-the-art (SOTA) attacks and 9 SOTA defense algorithms) and a standardized protocol of complete backdoor learning. We also provide comprehensive evaluations of every pair of 8 attacks against 9 defenses, with 5 poisoning ratios, based on 5 models and 4 datasets, thus 8,000 pairs of evaluations in total. We present abundant analysis from different perspectives about these 8,000 evaluations, studying the effects of different factors in backdoor learning. All codes and evaluations of BackdoorBench are publicly available at https://backdoorbench.github.io.

## 1   Introduction

With the widespread application of deep neural networks (DNNs) in many mission-critical scenarios, the security issues of DNNs have attracted more attentions. One of the typical security issue is backdoor learning, which could insert an imperceptible backdoor into the model through maliciously manipulating the training data or controlling the training process. It brings in severe threat to the widely adopted paradigm that people often download a unverified dataset/checkpoint to train/fine-tune their models, or even outsource the training process to the third-party training platform.

Although backdoor learning is a young topic in the machine learning community, its development speed is remarkable and has shown the state of a rapid arms race. When a new backdoor attack or defense method is developed based on an assumption or observation, it will be quickly defeated or evaded by more advanced adaptive defense or attack methods which break previous assumptions or observations. However, we find that the evaluations of new methods are often insufficient, with comparisons with limited previous methods, based on limited models and datasets. The possible reasons include the rapid development of new methods, diverse settings (*e.g.*, different threat models), as well as the difficulties of implementing or reproducing previous methods. Without thorough

---

*Corresponds to Baoyuan Wu (wubaoyuan@cuhk.edu.cn).

36th Conference on Neural Information Processing Systems (NeurIPS 2022) Track on Datasets and Benchmarks.

evaluations and fair comparisons, it is difficult to verify the real performance of a new method, as well as the correctness or generalization of the assumption or observation it is built upon. Consequently, we cannot well measure the actual progress of backdoor learning by simply tracking new methods. This dilemma may not only postpone the development of more advanced methods, but also preclude the exploration of the intrinsic reason/property of backdoor learning.

To alleviate this dilemma, we build a comprehensive benchmark of backdoor learning, called **BackdoorBench**. It is built on an extensible modular based codebase, consisting of the attack module, the defense module, as well as the evaluation and analysis module. Until now, we have implemented 8 stat-of-the-art (SOTA) backdoor attack methods and 9 SOTA defense methods, and provided 5 analysis tools (*e.g.*, t-SNE, Shapley value, Grad-CAM, frequency saliency map and neuron activation). More methods and tools are continuously updated. Based on the codebase, to ensure fair and reproducible evaluations, we also provide a standardized protocol of the complete procedure of backdoor learning, covering every step of the data preparation, backdoor attack, backdoor defense, as well as the saving, evaluation and analysis of immediate/final outputs. Moreover, we conduct comprehensive evaluations of every pair of attack and defense method (*i.e.*, 8 attacks against 9 defenses), with 5 poisoning ratios, based on 5 DNN models and 4 databases, thus up to 8,000 pairs of evaluations in total. These evaluations allow us to analyze some characteristics of backdoor learning. In this work, we present the analysis from four perspectives, to study the effects of attack/defense methods, poisoning ratios, datasets and model architectures, respectively. We hope that BackdoorBench could provide useful tools to facilitate not only the design of new attack/defense methods, but also the exploration of intrinsic properties and reasons of backdoor learning, such that to promote the development of backdoor learning.

Our main contributions are three-fold. **1) Codebase**: We build an extensible modular based codebase, including the implementations of 8 backdoor attack methods and 9 backdoor defense methods. **2) 8,000 comprehensive evaluations**: We provide evaluations of all pairs of 8 attacks against 9 defense methods, with 5 poisoning ratios, based on 4 datasets and 5 models, up to 8,000 pairs of evaluations in total. **3) Thorough analysis and new findings**: We present thorough analysis of above evaluations from different perspectives to study the effects of different factors in backdoor learning, with the help of 5 analysis tools, and show some interesting findings to inspire future research directions.

## 2 Related work

**Backdoor attacks** According to the threat model, existing backdoor attack methods can be partitioned into two general categories, including **data poisoning** and **training controllable**. **1) Data poisoning attack** means that the attacker can only manipulate the training data. Existing methods of this category focuses on designing different kinds of triggers to improve the imperceptibility and attack effect, including visible (*e.g.*, BadNets [15]) vs invisible (*i.e.*, Blended [5], Refool [31], Invisible backdoor [26]) triggers, local (*e.g.*, label consistent attack [44, 55]) vs global (*e.g.*, SIG [2]) triggers, additive (*e.g.*, Blended [5]) vs non-additive triggers (*e.g.*, smooth low frequency (LF) trigger [54], FaceHack [42]), sample agnostic (*e.g.*, BadNets [15]) vs sample specific (*e.g.*, SSBA [29], sleeper agent [46]) triggers, *etc.* The definitions of these triggers can be found in the bottom notes of Table 1. **2) Training controllable attack** means that the attacker can control both the training process and training data simultaneously. Consequently, the attacker can learn the trigger and the model weights jointly, such as LIRA [9], blind backdoor [1], WB [8], Input-aware [35], WaNet [36], *etc.*

**Backdoor defenses** According to the defense stage in the training procedure, existing defense methods can be partitioned into three categories, including **pre-training**, **in-training** and **post-training**. **1) Pre-training defense** means that the defender aims to remove or break the poisoned samples before training. For example, input anomaly detection and input pre-processing were proposed in [33] to block the backdoor activation by poisoned samples. Februus [7] firstly identified the location of trigger using Grad-CAM [43], and then used a GAN-based inpainting method [21] to reconstruct that region to break the trigger. NEO [50] proposed to use the dominant color in the image to generate a patch to cover the identified trigger. Confoc [51] proposed to change the style of the input image [12] to break the trigger. **2) In-training defense** means that the defender aims to inhibit the backdoor injection during the training. For example, anti-backdoor learning (ABL) [28] utilized the fact that poisoned samples are fitted faster than clean samples, such that they can be distinguished by the loss values in early learning epochs, then the identified poisoned samples are unlearned to mitigate the backdoor effect. DBD [20] observed that poisoned samples will gather together in the feature space

of the backdoored model. To prevent such gathering, DBD utilized the self-supervised learning [4] to learn the model backbone, then identified the poisoned samples according to the loss values when learning the classifier. **3) Post-training defense** means that the defender aims to remove or mitigate the backdoor effect from a backdoored model, and most existing defense methods belong to this category. They are often motivated by a property or observation of the backdoored model using some existing backdoor attacks. For example, the fine-pruning (FP) defense [30] and the neural attention distillation (NAD) [27] observed that poisoned and clean samples have different activation paths in the backdoored model. Thus, they aimed to mitigate the backdoor effect by pruning the neurons highly related to the backdoor. The channel Lipschitzness based pruning (CLP) method [56] found that the backdoor related channels often have a higher Lipschitz constant compared to other channels, such that the channels with high Lipschitz constant could be pruned to remove the backdoor. The activation clustering (AC) method [3] observed that samples of the target class will form two clusters in the feature space of a backdoored model, and the smaller cluster corresponds to poisoned samples. The spectral signatures (Spectral) method [49] observed that the feature representation distributions of poisoned and clean samples in the same class class are spectrally separable. The neural cleanse (NC) method [52] assumed that the trigger provides a "shortcut" between the samples from different source classes and the target class. The adversarial neuron pruning (ANP) defense [53] found that the neurons related to the injected backdoor are more sensitive to adversarial neuron perturbation (*i.e.*, perturbing the neuron weight to achieve adversarial attack) than other neurons in a backdoored model. We refer the readers to some backdoor surveys [11, 32] for more backdoor attack and defense methods.

**Related benchmarks** Several libraries or benchmarks have been proposed for evaluating the adversarial robustness of DNNs, such as CleverHans [39], Foolbox [40, 41], AdvBox [14], RobustBench [6], RobustART [48], ARES [10], Adversarial Robustness Toolbox (ART) [37], *etc.* However, these benchmarks mainly focused on adversarial examples [13, 24], which occur in the testing stage. In contrast, there are only a few libraries or benchmarks for backdoor learning (*e.g.*, TrojAI [22] and TrojanZoo [38]). Specifically, the most similar benchmark is TrojanZoo, which implemented 8 backdoor attack methods and 14 backdoor defense methods. However, there are significant differences between TrojanZoo and our BackdoorBench in two main aspects. **1) Codebase**: although both benchmarks adopt the modular design to ensure easy extensibility, TrojanZoo adopts the object-oriented programming (OOP) style, where each module is defined as one class. In contrast, BackdoorBench adopts the procedural oriented programming (POP) style, where each module is defined as one function, and each specific algorithm is implemented by several functions in a streamline. **2) Analysis and findings**. TrojanZoo has provided very abundant and diverse analysis of backdoor learning, mainly including the attack effects of trigger size, trigger transparency, data complexity, backdoor transferability to downstream tasks, and the defense effects of the tradeoff between robustness and utility, the tradeoff between detection accuracy and recovery capability, the impact of trigger definition. In contrast, BackdoorBench provides several new analysis from different perspectives, mainly including the effects of poisoning ratios and number of classes, the quick learning of backdoor, trigger generalization, memorization and forgetting of poisoned samples, as well as several analysis tools. In summary, we believe that BackdoorBench could provide new contributions to the backdoor learning community, and the competition among different benchmarks is beneficial to the development of this topic.

## 3 Our benchmark

### 3.1 Implemented algorithms

We have implemented 8 backdoor attack and 9 backdoor defense algorithms as the first batch of algorithms in our benchmark. We hold two criteria for choosing methods. **First**, it should be classic (*e.g.*, BadNets) or advanced method (*i.e.*, published in recent top-tier conferences/journals in machine learning or security community). The classic method serves as the baseline, while the advanced method represents the state-of-the-art, and their difference could measure the progress of this field. **Second**, the method should be easily implemented and reproducible. We find that some existing methods involve several steps, and some steps depend on a third-party algorithm or a heuristic strategy. Consequently, these methods involve too many hyper-parameters and are full of uncertainty, causing the difficulty on implementation and reproduction. Such methods are not included in BackdoorBench.

As shown in Table 1, the eight implemented backdoor attack methods cover two mainstream threat models, and with diverse triggers. Among them, BadNets[15], Blended[5] and LC[44] (label

Table 1: Categorizations of 8 backdoor attack algorithms in BackdoorBench, according to *threat models* and *different kinds of trigger characteristics*.

| Attack algorithm | Threat model | | Trigger characteristics | | | | | | | |
|---|---|---|---|---|---|---|---|---|---|---|
| | D-P | T-C | V | In-V | Local | Global | Add | N-Add | Ag | Sp |
| BadNets [15] | ✓ | | ✓ | | ✓ | | ✓ | | ✓ | |
| Blended [5] | ✓ | | | ✓ | | ✓ | ✓ | | ✓ | |
| LC [44] | ✓ | | ✓ | | ✓ | | ✓ | | ✓ | |
| SIG [2] | ✓ | | | ✓ | | ✓ | | ✓ | ✓ | |
| LF [54] | ✓ | | | ✓ | | ✓ | | ✓ | | ✓ |
| SSBA [29] | ✓ | | | ✓ | | ✓ | | ✓ | | ✓ |
| Input-aware [35] | | ✓ | ✓ | | ✓ | | ✓ | | | ✓ |
| WaNet [36] | | ✓ | | ✓ | | ✓ | | ✓ | | ✓ |

a) **Threat model**: D-P → data poisoning, *i.e.*, the attacker can only manipulate the training data;
   T-C → training controllable, *i.e.*, the attacker can control the training process and data;
b) **Trigger characteristics**:
   **b.1) Trigger visibility**: V → visible; In-V → invisible;
   **b.2) Trigger coverage**: Local → the trigger is a local patch; Global → the trigger covers the whole sample;
   **b.3) Trigger fusion mode**: Add → additive, *i.e.*, the fusion between the clean sample and the trigger is additive;
   N-Add → non-additive, *i.e.*, the fusion between the clean sample and the trigger is non-additive;
   **b.4) Trigger fusion mode**: Ag → agnostic, *i.e.*, the triggers in all poisoned samples are same;
   Sp → specific, *i.e.*, different poisoned samples have different triggers.

Table 2: Categorizations of 9 backdoor defense algorithms in BackdoorBench, according to four perspectives, including *input*, *output*, *defense stage* and *defense strategy*.

| Defense algorithm | Input | | | Output | | Defense stage | | Defense strategy | Motivation/Assumption/Observation |
|---|---|---|---|---|---|---|---|---|---|
| | B-M | S-CD | P-D | S-M | C-D | In-T | Post-T | | |
| FT | ✓ | ✓ | | ✓ | | | ✓ | 5 | Fine-tuning on clean data could mitigate the backdoor effect |
| FP [30] | ✓ | ✓ | | ✓ | | | ✓ | 2 + 5 | Poisoned and clean samples have different activation paths |
| NAD [27] | ✓ | ✓ | | ✓ | | | ✓ | 5 | Fine-tuning on clean data could mitigate the backdoor effect |
| NC [52] | ✓ | ✓ | | ✓ | | | ✓ | 1 + 4 + 5 | Trigger can be reversed through searching a shortcut to the target class |
| ANP [53] | ✓ | ✓ | | ✓ | | | ✓ | 2 + 5 | The backdoor related neurons are sensitive to adversarial neuron perturbation |
| AC [3] | | | ✓ | ✓ | ✓ | | ✓ | 3 + 5 | Samples labeled the target class will form 2 clusters in the feature space of a backdoored model |
| Spectral [49] | | | ✓ | ✓ | ✓ | | ✓ | 3 + 5 | The feature representations of poisoned and clean samples have different spectral signatures |
| ABL [28] | | | ✓ | ✓ | ✓ | ✓ | | 3 + 5 | Poisoned samples are learned more quickly than clean samples during the training |
| DBD [20] | | | ✓ | ✓ | ✓ | ✓ | | 3 + 6 | Poisoned samples will gather together in the feature space due to the standard supervised learning |

a) **Input**: B-M → a backdoored model; S-CD → a subset of clean samples; P-D → a poisoned dataset;
b) **Output**: S-M → secure model; C-D → clean data, *i.e.*, the subset of clean samples in the input poisoned data;
c) **Defense stage**: In-T → in-training, *i.e.*, defense happens during the training process;
   Post-T → post-training, *i.e.*, defense happens after the backdoor has been inserted through training;
d) **Defense strategy**: **1** → **backdoor detection**, *i.e.*, determining a model to be backdoored or clean;
   **2** → **backdoor identification**, *i.e.*, identifying the neurons in a backdoored model related to the backdoor;
   **3** → **poison detection**, *i.e.*, detecting poisoned samples;
   **4** → **trigger identification**, *i.e.*, identifying the trigger location in a poisoned sample;
   **5** → **backdoor mitigation**, *i.e.*, mitigating the backdoor effect of a backdoored model;
   **6** → **backdoor inhibition**, *i.e.*, inhibiting the backdoor insertion into the model during the training.

consistent attack) are three classic attack methods, while the remaining 5 are recently published methods. The general idea of each method will be presented in the **Supplementary Material**.

The basic characteristics of 9 implemented backdoor defense methods are summarized in Table 2, covering different inputs and outputs, different happening stages, different defense strategies. The motivation/assumption/observation behind each defense method is also briefly described in the last column. More detailed descriptions will be presented in the **Supplementary Material**.

## 3.2 Codebase

We have built an extensible modular-based codebase as the basis of BackdoorBench. As shown in Fig. 1, it consists of four modules, including *input module* (providing clean data and model architectures), *attack module*, *defense module* and *evaluation and analysis module*.

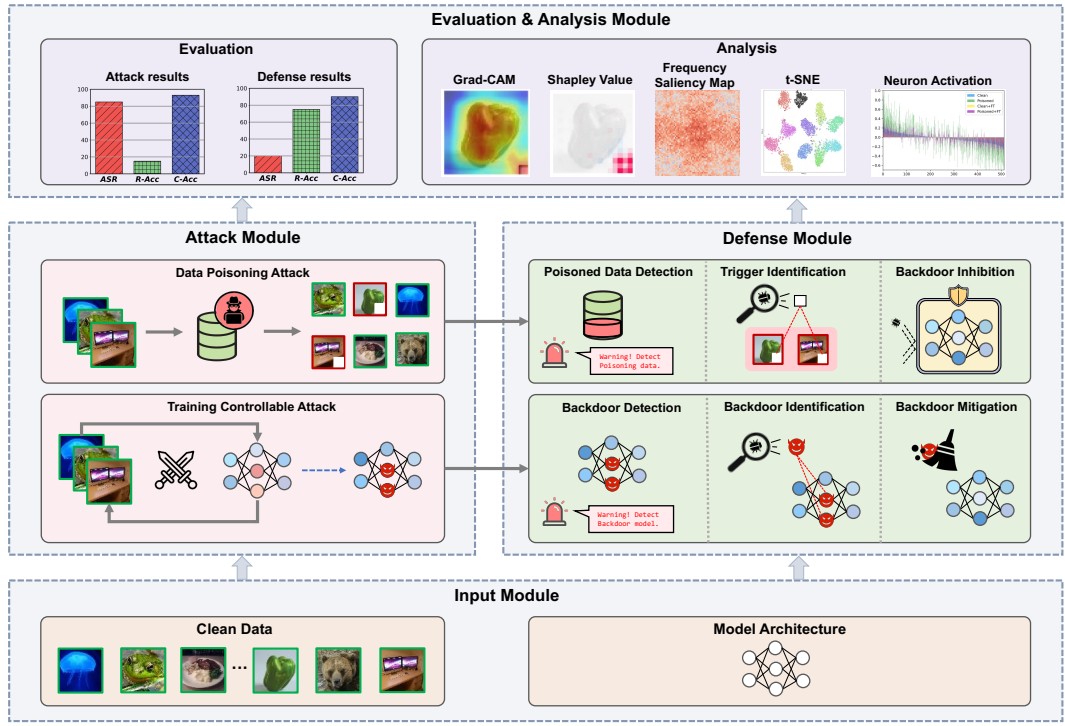

Figure 1: The general structure of the modular based codebase of BackdoorBench.

**Attack module** In the attack module, we provide two sub-modules to implement attacks of two threat models, $i.e.$, *data poisoning* and *training controllable* (see Table 1), respectively. For the first sub-module, it provides some functions of manipulating the provided set of clean samples, including trigger generation, poisoned sample generation ($i.e.$, inserting the trigger into the clean sample), and label changing. It outputs a poisoned dataset with both poisoned and clean samples. For the second sub-module, given a set of clean samples and a model architecture, it provides two functions of learning the trigger and model parameters, and outputs a backdoored model and the learned trigger.

**Defense module** According to the outputs produced by the attack module, there are also two sub-modules to implement backdoor defenses. If given a poisoned dataset, the first sub-module provides three functions of *poisoned sample detection* ($i.e.$, determining whether a sample is poisoned or clean), *trigger identification* ($i.e.$, identifying the location in the poisoned sample), *backdoor inhibition* ($i.e.$, training a secure model through inhibiting the backdoor injection). If given a backdoored model, as well as a small subset of clean samples (which is widely required in many defense methods), the second sub-module provides three functions of *backdoor detection* ($i.e.$, determining whether a model has a backdoor or not), *badckdoor identification* ($i.e.$, identifying the neurons in the backdoored model that are related to the backdoor effect), *backdoor mitigation* ($i.e.$, mitigating the backdoor effect from the backdoored model).

**Evaluation and analysis module 1)** We provide **three evaluation metrics**, including *clean accuracy (C-Acc)* ($i.e.$, the prediction accuracy of clean samples), *attack success rate (ASR)* ($i.e.$, the prediction accuracy of poisoned samples to the target class), *robust accuracy (R-Acc)* ($i.e.$, the prediction accuracy of poisoned samples to the original class). Note that the new metric R-Acc satisfies that ASR + R-Acc $\leq$ 1, and lower ASR and higher R-Acc indicate better defense performance. **2)** Moreover, we provide **five analysis tools** to facilitate the analysis and understanding of backdoor learning. *t-SNE* provides a global visualization of feature representations of a set of samples in a model, and it can help us to observe whether the backdoor is formed or not. *Gradient-weighted class activation mapping (Grad-CAM)* [43] and *Shapley value map* [34] are two individual analysis tools to visualize the contributions of different pixels of one image in a model, and they can show that whether the trigger activates the backdoor or not. We also propose the *frequency saliency map* to visualize the contribution of each individual frequency spectrum to the prediction, providing a novel perspective of backdoor from the frequency space. The definition will be presented in **Supplementary Material**.

*Neuron activation* calculates the average activation of each neuron in a layer for a batch of samples. It can be used to analyze the activation path of poisoned and clean samples, as well as the activation changes *w.r.t.* the model weights' changes due to attack or defense, providing deeper insight behind the backdoor.

**Protocol** We present a standardized protocol to call above functional modules to conduct fair and reproducible backdoor learning evaluations, covering every stage from data pre-processing, backdoor attack, backdoor defense, result evaluation and analysis, *etc.* We also provide three flexible calling modes, including *pure attack mode* (only calling an attack method), *pure defense mode* (only calling a defense method), as well as *a joint attack and defense mode* (calling an attack against a defense).

## 4 Evaluations and analysis

### 4.1 Experimental setup

**Datasets and models** We evaluate our benchmark on 4 commonly used datasets (CIFAR-10 [23], CIFAR-100 [23], GTSRB [17], Tiny ImageNet [25]) and 5 backbone models (PreAct-ResNet18[2] [16], VGG-19 [3] [45] (without the batchnorm layer), EfficientNet-B3 [4] [47], MobileNetV3-Large [5][18], DenseNet-161 [6] [19]). To fairly measure the performance effects of the attack and defense method for each model, we only used the basic version of training for each model without adding any other training tricks (*e.g.*, augmentation). The details of datasets and clean accuracy [7] of normal training are summarized in Table 3.

Table 3: Dataset details and clean accuracy of normal training.

| Datasets | Classes | Training/ Testing Size | Image Size | Clean Accuracy | | | | |
|---|---|---|---|---|---|---|---|---|
| | | | | PreAct-ResNet18 [16] | VGG-19 [45] | EfficientNet-B3 [47] | MobileNetV3-Large [18] | DenseNet-161[19] |
| CIFAR-10 [23] | 10 | 50,000/10,000 | $32 \times 32$ | 93.90% | 91.38% | 64.69% | 84.44% | 86.82% |
| CIFAR-100 [23] | 100 | 50,000/10,000 | $64 \times 64$ | 70.51% | 60.21% | 48.92% | 50.73% | 57.57% |
| GTSRB [17] | 43 | 39,209/12,630 | $32 \times 32$ | 98.46% | 95.84% | 87.39% | 93.99% | 92.49% |
| Tiny ImageNet [25] | 200 | 100,000/10,000 | $64 \times 64$ | 57.28% | 46.13% | 41.08% | 38.78% | 51.73% |

**Attacks and defenses** We evaluate each pair of 8 attacks against 9 defenses in each setting, as well as one attack without defense. Thus, there are $8 \times (9 + 1) = 80$ pairs of evaluations. We consider 5 poisoning ratios, *i.e.*, $0.1\%, 0.5\%, 1\%, 5\%, 10\%$ for each pair, based on all 4 datasets and 5 models, leading to $8,000$ pairs of evaluations in total. The performance of every model is measured by the metrics, *i.e.*, C-Acc, ASR and R-Acc (see Section 3.2). The implementation details of all algorithms, and the results of the DBD defense [20] will be presented in the **Supplementary Material**.

### 4.2 Results overview

We first show the performance distribution of various attack-defense pairs under one model structure (*i.e.*, PreAct-ResNet18) and one poisoning ratio (*i.e.*, $5\%$) in Figure 2. In the top row, the performance is measured by clean accuracy (C-Acc) and attack success rate (ASR). From the attacker's perspective, the perfect performance should be high C-Acc and high ASR simultaneously, *i.e.*, located at the top-right corner. From the defender's perspective, the performance should be high C-Acc and low ASR simultaneously, *i.e.*, located at the top-left corner. It is observed that most color patterns locate at similar horizontal levels, reflecting that most defense methods could mitigate the backdoor effect while not harming the clean accuracy significantly. In the bottom row, the performance is measured by robust accuracy (R-Acc) and ASR. As demonstrated in Section 3.2, ASR + R-Acc $\leq 1$. From the defender's perspective, it is desired that the reduced ASR value equals to the increased R-Acc, *i.e.*, the prediction of the poisoned sample is recovered to the correct class after the defense. It is interesting to see that most color patterns are close to the anti-diagonal line (*i.e.*, ASR + R-Acc

---

[2]https://github.com/VinAIResearch/Warping-based_Backdoor_Attack-release/blob/main/classifier_models/preact_resnet.py

[3]https://pytorch.org/vision/0.12/_modules/torchvision/models/vgg.html#vgg19

[4]https://pytorch.org/vision/main/_modules/torchvision/models/efficientnet.htmlefficientnet_b3

[5]https://github.com/pytorch/vision/blob/main/torchvision/models/mobilenetv3.py

[6]https://pytorch.org/vision/main/_modules/torchvision/models/densenet.htmldensenet161

[7]Note that to fairly measure the effects of the attack and defense method, we train all victim models from scratch without further training tricks, which explains the low clean accuracy of some models.

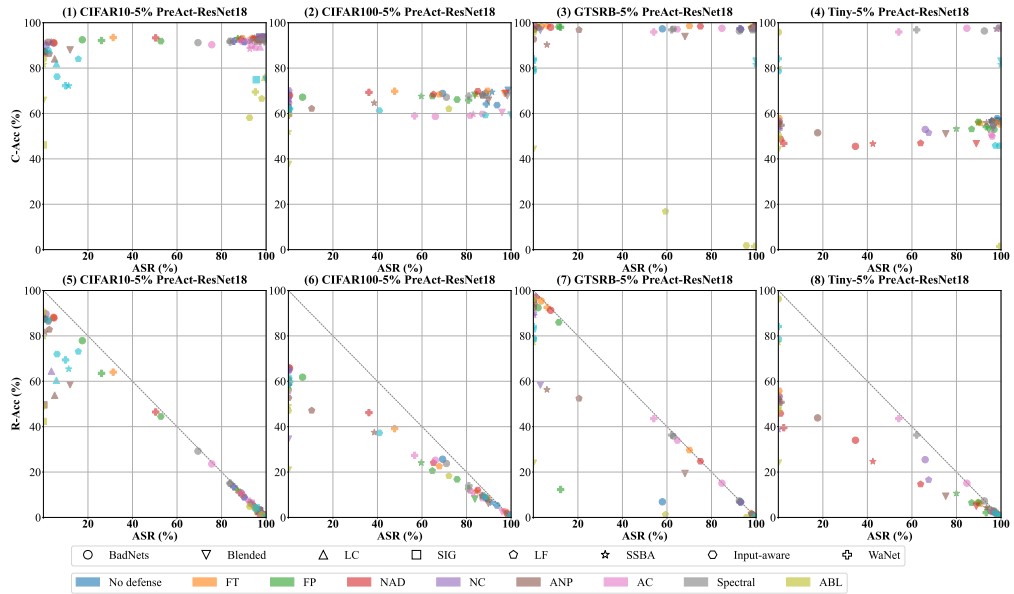

Figure 2: Performance distribution of different attack-defense pairs. Each color pattern represents one attack-defense pair, with attacks distinguished by patterns, while defenses by colors.

= 1) on CIFAR-10 (the first column) and GTSRB (the third column), while most patterns are from that line on CIFAR-100 (the second column) and Tiny ImageNet (the last column). We believe it is highly related to the number of classes of the dataset. Given a large number of classes, it is more difficult to recover the correct prediction after the defense. These figures could provide a big picture of the performance of most attacks against defense methods. Due to the space limit, the results of other settings will be presented in the **Supplementary Material**.

### 4.3 Effect of poisoning ratio

Here we study the effect of the poisoning ratio on the backdoor performance. Figure 3 visualizes the results on CIFAR-10 and PreAct-ResNet18, *w.r.t.* each poisoning ratio for all attack-defense pairs, and each sub-figure corresponds to each defense. In sub-figures (1,6,7), ASR curves increase in most cases, being consistent with our initial impression that higher poisoning ratios lead to stronger attack performance. However, in other sub-figures, there are surprisingly sharp drops in ASR curves. To understand such *abnormal* phenomenon, we conduct deep analysis for these defenses, as follows.

**Analysis of FT/FP/NAD/NC**   The curves for FT, FP [30], NAD [27] (its plots will be presented in **Supplementary Material**) and NC[52] are similar since they all use fine-tuning on a small subset of clean data (*i.e.*, 5% training data), thus we present a deep analysis for FT as an example. As shown Figure 4, we compare the performance of 5% and 10%. We first analyze the changes in the average neuron activation (see Section 3.2) before and after the defense. As shown in the top row, the changes between *Poisoned+No Defense* (green) and *Poisoned+FT* (purple) in the case of 5% are much smaller than those in the case of 10%. It tells that the backdoor is significantly affected by FT. We believe the reason is that when the poisoning ratio is not very high (*e.g.*, 5%), the model fits clean samples very well, while the fitting gets worse if the poisoning ratio keeps increasing after a threshold ratio. We find that the clean accuracy on the 5% clean data used for fine-tuning by the backdoored model before the defense is 99% in the case of 5% poisoning ratio, while 92% in the case of 10% poisoning ratio. It explains why their changes in neuron activation values are different.

**Analysis of ABL**   The ABL [28] method uses the loss gap between the poisoned and clean samples in the early training period to isolate some poisoned samples. We find that the loss gap in the case of high poisoning ratio is larger than that in the case of low poisoning ratio. Take the LC [44] attack on CIFAR-10 as example. In the case of 5% poisoning ratio, the isolated 500 samples by ABL are 0 poisoned and 500 clean samples, such that the backdoor effect cannot be mitigated in later backdoor unlearning in ABL. In contrast, the isolated 500 samples are all poisoned in the case of 10% poisoning ratio. The t-SNE visualizations shown in the second row of Figure 4 also verify this point.

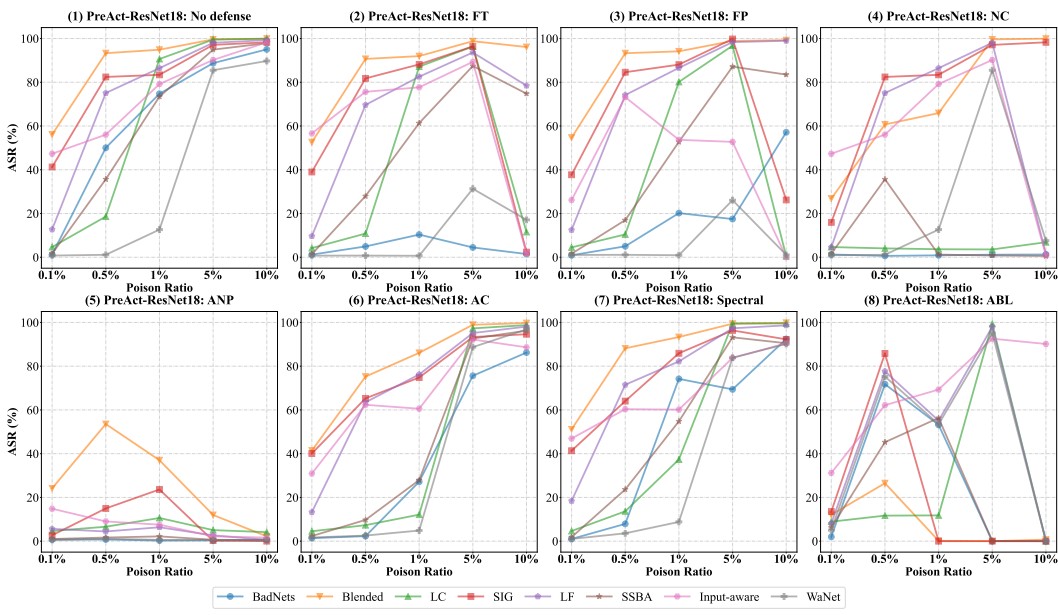

Figure 3: The effects of different poisoning ratios on backdoor learning.

**Analysis of ANP** The ANP [53] prunes the neurons that are sensitive to the adversarial neuron perturbation, by setting a threshold. As suggested in [53], this threshold is fixed as 0.2 in our evaluations. We find that when the poisoning ratio is high, more neurons will be pruned, thus the ASR may decrease. For example, given the SIG [2] attack, the pruned neurons by ANP are 328 and 466 for $5\%$ and $10\%$ poisoning ratios, respectively. As shown in the last row of Figure 4, poisoned samples still gather together for $5\%$, while separated for $10\%$.

**In summary**, the above analysis demonstrates an interesting point that attack with higher poisoning ratios doesn't mean better attack performance, and it may be more easily defended by some defense methods. The reason is that higher poisoning ratios will highlight the difference between poisoned and clean samples, which will be utilized by adaptive defenses. This point inspires two interesting questions that deserve further exploration in the future: *how to achieve the desired attack performance using fewer poisoned samples, and how to defend weak attacks with low poisoning ratios*. Moreover, considering the randomness due to weight initialization and some methods' mechanisms, we repeat the above evaluations several times. Although some fluctuations occur, the trend of ASR curves is similar to that in Figure 3. More details and analysis are presented in **Supplementary Material**.

## 4.4 Effect of model architectures

As shown in Figure 5, we analyze the influence caused by model architectures. From the top-left sub-figure, it is worth noting that, under the same training scheme, not all backdoor attacks can successfully plant a backdoor in EfficientNet-B3, such as BadNets, LC, SSBA, and WaNet. In contrast, PreAct-ResNet18 is easy to be planted a backdoor. Besides, we find that most defense methods fail to remove the backdoors embedded in the PreAct-ResNet18 and VGG-19, except ANP. However, ANP is less effective on EfficientNet-B3 attacked by SIG. From the second sub-figure in the first row, we notice that FT is an optimal defense method for MobileNetV3-Large, which could effectively decrease the ASR. In most cases, NC and ANP can remove the backdoors embedded in DenseNet-161. The above analysis demonstrates that one attack or defense method may have totally different performance on different model architectures. It inspires us *to further study the effect of model architecture in backdoor learning and to design more robust architectures in the future*.

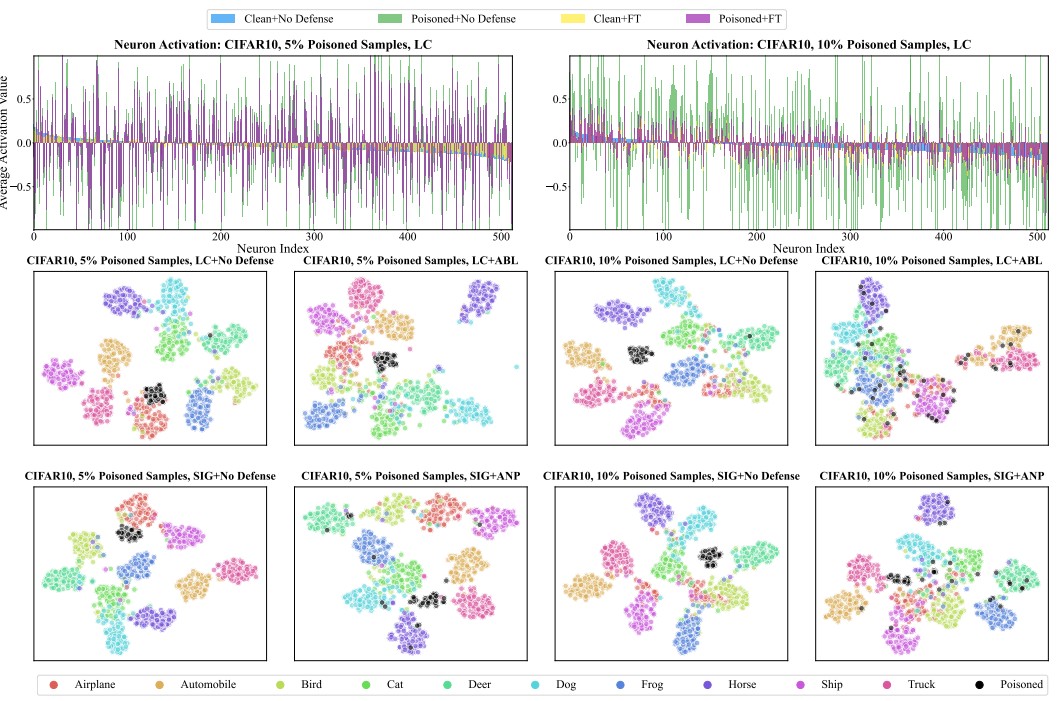

Figure 4: The changes of neuron activation values due to the FT defense (**Top row**), and the changes of t-SNE visualization of feature representations due to the ABL defense (**Middle row**) and the ANP defense (**Bottom row**), respectively.

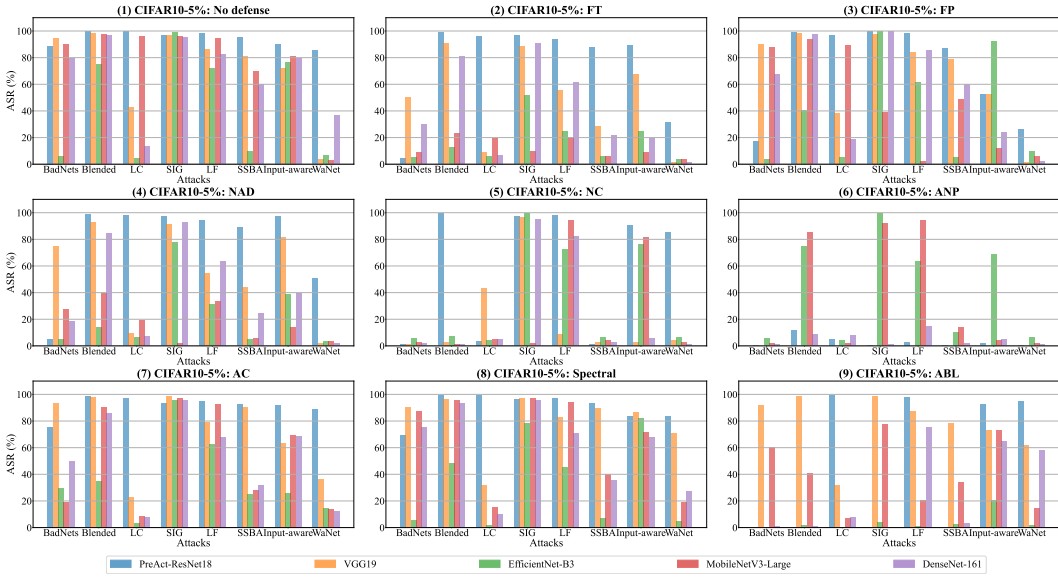

Figure 5: The effects of different model architectures using different defense and attack methods.

## 4.5   Contents in supplementary material

Due to the space limit, we have put several important contents in the **Supplementary Material**. Here we present a brief outline of the Supplementary Material to facilitate readers to find the corresponding content, as follows:

- **Section A:** Additional information of backdoor attack and defense algorithms:

- Section A.1: Descriptions of backdoor attack algorithm;
- Section A.2: Descriptions of backdoor defense algorithms;
- Section A.3: Implementation details and computational complexities.

- **Section B:** Additional evaluations and analysis:
  - Section B.1: Full results on CIFAR-10;
  - Section B.2: Results overview;
  - Section B.3: Effect of dataset;
  - Section B.4: Effect of poisoning ratio;
  - Section B.5: Sensitivity to hyper-parameters;
  - Section B.6: Analysis of quick learning of backdoor;
  - Section B.7: Analysis of backdoor forgetting;
  - Section B.8: Analysis of trigger generalization of backdoor attacks;
  - Section B.9: Evaluation on vision transformer;
  - Section B.10: Evaluation on ImageNet;
  - Section B.11: Visualization.

- **Section C:** BackdoorBench in Natural Language Processing;
- **Section D:** Reproducibility;
- **Section E:** License.

## 5 Conclusions, limitations and societal impacts

**Conclusions** We have built a comprehensive and latest benchmark for backdoor learning, including an extensible modular-based codebase with implementations of 8 advanced backdoor attacks and 9 advanced backdoor defense algorithms, as well as 8,000 attack-defense pairs of evaluations and thorough analysis. We hope that this new benchmark could contribute to the backdoor community in several aspects: providing a clear picture of the current progress of backdoor learning, facilitating researchers to quickly compare with existing methods when developing new methods, and inspiring new research problems from the thorough analysis of the comprehensive evaluations.

**Limitations** Until now, BackdoorBench has mainly provided algorithms and evaluations in the computer vision domain and supervised learning. In the future, we plan to expand BackdoorBench to more domains and learning paradigms, $e.g.$, natural language processing (NLP), Speech, and reinforcement learning.

**Societal impacts** Our benchmark could facilitate the development of new backdoor learning algorithms. Meanwhile, like most other technologies, the implementations of backdoor learning algorithms may be used by users for good or malicious purposes. The feasible approach to alleviate or avoid adverse impacts could be exploring the intrinsic property of the technology, regulations, and laws.

## 6 Acknowledgement

This work is supported by the National Natural Science Foundation of China under grant No.62076213, Shenzhen Science and Technology Program under grant No.RCYX20210609103057050, and the university development fund of the Chinese University of Hong Kong, Shenzhen under grant No.01001810. Chao Shen is supported by the National Key Research and Development Program of China (2020AAA0107702), National Natural Science Foundation of China (U21B2018, 62161160337, 62132011), Shaanxi Province Key Industry Innovation Program (2021ZDLGY01-02).

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
