# OpenReview forum: "BackdoorBench: A Comprehensive Benchmark of Backdoor Learning"
_NeurIPS.cc/2022/Track/Datasets_and_Benchmarks — NeurIPS 2022 Datasets and Benchmarks _

### Official Review · Reviewer_rrL4 · 2022-06-30

**Rating:** 7
**Confidence:** 3
**Correctness:** Yes.

**Strengths:**

The paper tackles a relatively new and important subfield of research, attacking/defending backdoor attacks in deep neural networks.

The authors provide implementations of the latest backdoor attack and defense algorithms.

The evaluation is relatively comprehensive and provides some insights on the behavior of the attack and defense methods under different conditions.

The paper is well structured and easy to read and follow.

The authors provide a nice overview and general taxonomy of the different attack and defense methods.

The codebase is extensible to more attacks/defenses/models.

**Weaknesses:**

My main concern is the similarity of BackdoorBench to TrojanZoo. TrojanZoo also provides a codebase of 8 attacks and 14 defenses; however, the authors note some of these methods are older and out of date. Can't TrojanZoo be updated with the latest attacks/defenses? What is the main benefit of adding an additional backdoor benchmark? Is BackdoorBench more easily extensible than TrojanZoo? If BackdoorBench is not easily extensible, then a new backdoor benchmark codebase may be created again in 1-2 years, possibly hindering the comparison of new attacks/defenses to older methods.

The authors note the predictive performance of some of the models are relatively low due to differences in training. Analyzing backdoor attack/defense performance seems less useful when evaluating based on poor predictive models. How did the authors choose these 5 model architectures to evaluate? And are many of the attacks the authors implemented model agnostic?

Backdoor attack/defense performance is only evaluated on image classification datasets; however, I think including text or audio datasets could significantly improve this paper.

How robust are the results in Figure 3, where are the error bars?


MINOR WEAKNESSES

line 102: "class" repeated.

Text on Figures 2,3,4,5 is small and hard to read.

**Additional Feedback:**

Perhaps provide some results of the evaluation analysis in the abstract or introduction.

**Clarity:**

The paper is well structured, but contains numerous grammatical errors. I recommend using a service like Grammarly to fix these issues.

**Documentation:**

Yes.

**Ethics:**

No.

**Relation To Prior Work:**

Yes.

**Summary And Contributions:**

The main contribution of the paper is a codebase that implements the latest attack, defense, and evaluation methods related to backdoor attacks on deep learning models; in total, the authors implemented 8 state-of-the-art (SOTA) attack algorithms and 9 SOTA defense algorithms.

The second and third contributions are an evaluation and analysis of the 8 attacks against the 9 defenses using 5 poisoning ratios (0.1%, 0.5%, 1%, 5%, 10%) based on 5 model architectures trained on 4 image classification datasets; a total of 8,000 evaluations were performed ("no defense" was added as an additional defense strategy in the evaluation).

Overall, the authors find that most defense methods can mitigate many of the backdoor attacks while not significantly harming accuracy on the set of clean examples. They also find defenses have a more difficult time recovering the correct prediction for poisoned examples when the dataset has a relatively large number of classes. Additionally, the authors find some defenses achieve a lower attack success rate when the poison ratio is higher, since a higher poison ratio may be easier for that defense method to detect. Finally, their analysis suggests that attack/defense performance can vary considerably depending on the model architecture, prompting further investigation into the relationship between model architecture and backdoor performance.

---

> ### Author Response · Authors · 2022-08-12
> **Response to Reviewer rrL4: greatly appreciate your constructive comments and high recognition about our efforts (Part 2)**
>
>
> **Q3: Backdoor attack/defense performance is only evaluated on image classification datasets; however, I think including text or audio datasets could significantly improve this paper.**
>
> **R3:** Thanks for this constructive suggestion. As you suggested, the extensions to adding text or audio datasets into the benchmark have been added into our future schedule (please refer to the last section of the common response to all reviewers posted at the top location). And, we are implementing and evaluating some backdoor attack and defense methods in the natural language processing (NLP) domain. Once the first batch of evaluations is finished, we will immediately post it here and add them to the revised manuscript. We will keep you informed about the progress.
>
>
> ---
>
> **Q4: How robust are the results in Figure 3, where are the error bars?**
>
> **R4:** Thanks for this constructive suggestion. We are running the evaluations several times with random initializations. We will update Figure 3 in the revised manuscript when these experiments are finished (maybe within 4 days, *i.e.*, before August 16, 2022). We will keep you informed about the progress.
>
> ---
>
> **Q5: The paper is well structured, but contains numerous grammatical errors. I recommend using a service like Grammarly to fix these issues.**
>
> **R5:** Greatly appreciate your careful inspection. We will seriously follow your suggestion to thoroughly proofread the manuscript to fix grammatical/spelling errors.
>
> ---
>
> **Q6: Text on Figures 2,3,4,5 is small and hard to read. Perhaps provide some results of the evaluation analysis in the abstract or introduction.**
>
> **R6:** Thanks for your constructive suggestions. We will follow these suggestions in the revised manuscript, and the revised manuscript will be uploaded after most updates have been finished.
>
> ---
>
> To avoid too long wait, we firstly post the above preliminary responses to your concerns. All additional results/analysis promised above should be immediately added here and in the revised manuscript, once they are finished. Hope these responses helpful to address your concerns. And, we are willing to discuss with you about any further concerns.
>
> Thanks again for your constructive comments and your recognition of our efforts.
>
> Sincerely,
>
> Authors

---

> ### Author Response · Authors · 2022-08-12
> **Response to Reviewer rrL4: greatly appreciate your constructive comments and high recognition about our efforts (Part 1)**
>
> Dear Reviewer rrL4,
>
> We sincerely appreciate your precious time and constructive comments, and are greatly encouraged by your high recognition about our efforts to build a new backdoor benchmark.
> In the following, we would like to answer your concerns separately.
>
> ---
>
> **Q1: My main concern is the similarity of BackdoorBench to TrojanZoo**. TrojanZoo also provides a codebase of 8 attacks and 14 defenses; however, the authors note some of these methods are older and out of date. Can't TrojanZoo be updated with the latest attacks/defenses? What is the main benefit of adding an additional backdoor benchmark? IsBackdoorBench more easily extensible than TrojanZoo? If BackdoorBench is not easily extensible, then a new backdoor benchmark codebase may be created again in 1-2 years, possibly hindering the comparison of new attacks/defenses to older methods.
>
> **R1:** Thanks for this insightful comment. We sincerely admire the efforts and contributions of all pioneer backdoor benchmarks, especially the great benchmark TrojanZoo. Actually we have learned a lot from them, and we always think about what new valuable contributions we can make to the backdoor community.
>
> We would like to invite you to see **the common response** named "**A common response to the concern about the difference between BackdoorBench and TrojanZoo**" posted at the top location for more details, where we demonstrate the significant differences in codebase, analysis and findings between our BackdoorBench and TrojanZoo, as well as our motivation, the latest progress and our future schedule. Hope those responses be helpful to evaluate the contribution of our benchmark to the backdoor community.
>
> ---
>
> **Q2: The authors note the predictive performance of some of the models are relatively low due to differences in training. Analyzing backdoor attack/defense performance seems less useful when evaluating based on poor predictive models. How did the authors choose these 5 model architectures to evaluate? And are many of the attacks the authors implemented model agnostic?**
>
> **R2:** Thanks for this insightful comment. We would like to explain from the following aspects.
>
> - **Model choice**: we try to cover diverse model architectures of different structures, different sizes, such that we can observe and analyze the effect of model architectures in the backdoor learning. These 5 model architectures are widely used, typical ones, thus we choose them in our evaluations.
>
> - **Is it really less useful of analyzing the backdoor attack/defense performance based on poor predictive models?** Your comment inspires us to deeply think about the correlation between the backdoor learning performance and the model's predictive capability. As shown in our evaluations, given the same dataset, one attack/defense method may performs significantly different on different models. It is supposed to be related with both the model architecture and the model's predictive capability. In some challenging applications and datasets, the best performance of deep models is still not very high. In this case, what will happen to the backdoor learning? Is it easier or more difficult to achieve the comparable attack/defense performance, compared with the backdoor learning on the model with very high predictive performance (like PreAct-ResNet18 on CIFAR-10). We think that there is good practical value to analyze the correlation between the backdoor learning performance and the model's predictive capability. We plan to conduct more in-depth analysis about this point in our benchmark in the future. Thanks again for your inspired comment.
>
> - "**Are many of the attacks the authors implemented model agnostic?**" Yes, as shown at the second column of Table 1 in the manuscript, the top six attack methods are data poisoning based method, meaning that they only manipulate the training data, without any control of the training process, including which model used in training. Thus, all of them are model agnostic.

---

> ### Author Response · Authors · 2022-08-19
> **Latest response to Reviewer rrL4: Thanks for your help, and all suggested updates have been added**
>
> Dear Reviewer rrL4,
>
> We sincerely appreciate your constructive comments and warming encouragements. All suggested updates by you and other reviewers have been added into the revised manuscript and the codebase of BackdoorBench, and are summarized in the latest common response posted above.
>
> Hope our responses and updates helpful for you to measure the value of our work. We are looking forward to your further feedback, and are willing to answer any further concern. Thanks.
>
> Best regards,
>
> Authors

---

### Official Review · Reviewer_C28j · 2022-07-23
**A computer vision backdoor learning benchmark**

**Rating:** 7
**Confidence:** 4
**Correctness:** The benchmark has appropriate experim…
**Clarity:** This paper is well written.

**Strengths:**

1. This paper provides different backdoor attack and defense methods in Computer Vision domain, this is good.
2. clear and runnable codes for individual attack/defense methods, the results table is very persuasive.

**Weaknesses:**

1. The frameworks are not well compatible. A very good point is: For individual attack/defense method, the framework provides seperate and individual script to make sure the code can run. However, there are many redundant codes, such the data loader part. If further work can reduce the redundance of the codes, and make the framework more light-weighted, it would be better.
2. What's the difference between your work and other benchmarks, such as 'backdoor101' and 'trojanzoo'. The paper mensioned in 'related benchmarks' part, claiming they contain recent models. But this is not very persuasive. However, compared to 'trojanzoo', your codebase seems easier to extend to other datasets.
3. In the future, if the benchmark can extend to NLP domain, it would be a huge jump and wonderful improvement.

**Additional Feedback:**

Nope

**Documentation:**

The documentation is good.

**Ethics:**

Nope

**Relation To Prior Work:**

yes.

**Summary And Contributions:**

This paper provides a Benchmark on Computer Vision Backdoor Learning, with an easy implementations of mainstream CV backdoor attack and defense methods. it contains 8 attack methods, 9 defense methods with 4 datasets, as well as the detailed results.

---

> ### Author Response · Authors · 2022-08-13
> **Response to Reviewer C28j: greatly appreciate your constructive comments and high recognition about our efforts**
>
> Dear Reviewer C28j,
>
> We sincerely appreciate your precious time and constructive comments, and are greatly encouraged by your high recognition about our efforts to build a new backdoor benchmark.
> In the following, we would like to answer your concerns separately.
>
> ---
>
> **Q1: Reducing code redundancy**: The frameworks are not well compatible. A very good point is: For individual attack/defense method, the framework provides seperate and individual script to make sure the code can run. However, there are manyredundant codes, such the data loader part. If further work can reduce the redundance of the codes, and makethe framework more light-weighted, it would be better.
>
> **R1:** Thanks for this constructive suggestion. Following your suggestion, we have merged several redundant codes of data loader into the `trainer` in the implementation of each attack method (please refer to the folder `https://github.com/SCLBD/BackdoorBench/tree/main/attack/` and the commit *585a8c2311fb136eda34499adaf9c4e59cc7dd91*). We will keep trying to improve the code quality through making the code more light-weighted while maintaining its flexibility and readability.
>
> ---
>
> **Q2: Difference with previous benchmarks**: What's the difference between your work and other benchmarks, such as 'backdoor101' and 'trojanzoo'. The paper mensioned in 'related benchmarks' part, claiming they contain recent models. But this is not very persuasive. However, compared to 'trojanzoo', your codebase seems easier to extend to other datasets.
>
> **R2:** Thanks for this insightful comment. We sincerely admire the efforts and contributions of all pioneer backdoor benchmarks. Actually we have learned a lot from them, and we always think about what new valuable contributions we can make to the backdoor community.
>
> We would like to invite you to see **the common response** named "**A common response to the concern about the diff erencebetween BackdoorBench and TrojanZoo**" posted at the top location for more details, where we demonstrate the significant diferences in codebase, analysis and findings between our BackdoorBench and TrojanZoo, as well as our motivation, the latest progress and our future schedule. Note that as 'backdoor101' have povided the implementations of only 3 attack and 3 defense metohds until now, without evaluations and analysis, our responses mainly focus on the TrojanZoo.
> Hope those responses be helpful to evaluate the contribution of our benchmark to the backdoor community.
>
> ---
>
>
> **Q3: Extensions to other domains**: In the future, if the benchmark can extend to NLP domain, it would be a huge jump and wonderfulimprovement.
>
> **R3:** Thanks for this constructive suggestion. As you suggested, the extensions to other domains, including NLP and Speech, have been added into our future schedule (please refer to the last section of the common response to all reviewers posted at the top location).
> And, we are implementing and evaluating some backdoor attack and defense methods in the NLP domain. Once the first batch of evaluations is finished, we will immediately post it here and add them to the revised manuscript. We will keep you informed about the progress.
>
> ---
>
> To avoid too long wait, we firstly post the above preliminary responses to your concerns. All additional code-update/results/analysis promised above will be immediately added here and in the revised manuscript, once they are finished. Hope these responses helpful to address your concerns. And, we are willing to discuss with you about any further concerns.
>
>
> Thanks again for your constructive comments and your recognition of our efforts.
>
> Sincerely,
>
> Authors

---

> ### Author Response · Authors · 2022-08-19
> **Latest response to Reviewer C28j: Thanks for your help, and all suggested updates have been added**
>
> Dear Reviewer C28j,
>
> We sincerely appreciate your constructive comments and warming encouragements. All suggested updates by you and other reviewers have been added into the revised manuscript and the codebase of BackdoorBench, and are summarized in the latest common response posted above.
>
> Hope our responses and updates helpful for you to measure the value of our work. We are looking forward to your further feedback, and are willing to answer any further concern. Thanks.
>
> Best regards,
>
> Authors

---

### Official Review · Reviewer_Aamn · 2022-07-26
**A good benchmark on backdoor attacks/defenses**

**Rating:** 8
**Confidence:** 5

**Additional Feedback:**

Please answer the reviewer's questions in [Weakness].

**Documentation:**

Yes.

**Ethics:**

N/A.

**Summary And Contributions:**

This paper proposes a comprehensive backdoor benchmark of backdoor learning. It consists of 8 state-of-the-art backdoor attacks and 9 backdoor defenses. Based on the open-sourced framework, the authors conducted large-scale experiments on 4 datasets and provided some preliminary analysis and findings on these attacks/defenses. Overall, this paper offers a very easy-to-use backdoor benchmark, which would be highly beneficial for researchers in the backdoor learning area.

---

> ### Author Response · Authors · 2022-08-13
> **Response to Reviewer Aamn: greatly appreciate your constructive comments and high recognition about our efforts**
>
> Dear Reviewer Aamn,
>
> We sincerely appreciate your precious time and constructive comments, and are greatly encouraged by your high recognition about our efforts to build a new backdoor benchmark.
> In the following, we would like to answer your concerns separately.
>
> ---
>
> **Q1: Sugestions of adding ImageNet and transformer-based architectures**: As for the experiments of this paper, I suggest the author also involve the ImageNet dataset in the benchmark in the future since the image size for current datasets (e.g., CIFAR, Tiny-ImageNet) is still relatively small; I suggest the author consider transformer-based architectures and evaluate their robustness on backdoorattacks;
>
> **R1:** Thanks for this constructive suggestion. Following your suggestion, we have added the ImageNet dataset and the ViT model into the codebase (please refer to Line 371 of `https://github.com/SCLBD/BackdoorBench/blob/main/utils/aggregate_block/dataset_and_transform_generate.py` and Line 84 of `https://github.com/SCLBD/BackdoorBench/blob/main/utils/aggregate_block/model_trainer_generate.py`, respectively). The evaluations on them are still in running.
>  Once the first batch of evaluations is finished, we will immediately post it here and add them to the revised manuscript. We will keep you informed about the progress.
>
> ---
>
> **Q2: The author should introduce and compare adversarial benchmarks RobustART, which also studies the robustness of architectures.**
>
> **R2:** Thanks for this constructive suggestion. RobustART is an amazing benchmark which
> provides the first comprehensive evaluation of the effect of model architectures and training techniques to the adversarial robustness.  Although the studied topic is different (RobustART studied the robustness to adversarial examples, while BackdoorBench studied the backdoor learning), the evaluation settings and analysis perspectives in RobustART could inspire us to conduct deeper evaluations and analysis about the effect of model architectures to the backdoor learning, as demonstrated in Section 4.4 of the main manuscript. The citation of RobustART has been added into the *Related  benchmarks* paragraph of Section 2 in the revised manuscript), and the studies about model architectures will be continuously added into our benchmark.
>
> **Q3: Minor issues: I suggest the author refine Table1 and Table2, which are hard for me to understand now.**
>
> **R3:** Thanks for for this constructive suggestion. We will refine the tables in the revision to   make them more clearly.
>
> ---
>
> **Q4: It seems that the author did not address the limitations and broader impacts in the main body of the paper, and I suggest the author add them in the revision.**
>
> **R4:** Thanks for your careful inspection. We will add the *limitations and broader impacts* into the revised main manuscript.
>
> ---
>
> To avoid too long wait, we firstly post the above preliminary responses to your concerns. All additional results/analysis promised above will be immediately added here and in the revised manuscript, once they are finished.
> The revised manuscript will be uploaded after finishing most changes.
> Hope these responses helpful to address your concerns. And, we are willing to discuss with you about any further concerns.
>
>
> Thanks again for your constructive comments and your recognition of our efforts.
>
> Sincerely,
>
> Authors

---

> > ### Comment · Reviewer_Aamn · 2022-08-16
> > **Post-rebuttal**
> >
> > The reviewer thanks the authors for the detailed responses, and most of my concerns have been addressed. After reading the responses and other reviewers' comments, I am very positive about the contribution of this paper to the community. Therefore, I raised my score.

---

> > > ### Author Response · Authors · 2022-08-19
> > > **Thanks for your help, and all suggested updates have been added**
> > >
> > > Dear Reviewer Aamn,
> > >
> > > We sincerely appreciate your constructive comments and warming encouragements. All suggested updates by you and other reviewers have been added into the revised manuscript and the codebase of BackdoorBench, and are summarized in the latest common response posted above.
> > >
> > > Best regards,
> > >
> > > Authors

---

### Official Review · Reviewer_RPfC · 2022-07-26
**Nice benchmark effort, but challenges in distinguishing itself from prior work**

**Rating:** 4
**Confidence:** 4

**Strengths:**

- Large number of defenses and attacks evaluated
- Code base that enables easy implementation of defenses and attacks
- Public benchmark on the defense performance

The proposed code base is well-documented and comprehensive in terms of including existing backdoor attacks and defenses. I appreciate authors' effort in running 8000 experiments on defense and attack pairs.

**Weaknesses:**

- There exists many code bases on backdoor attacks.
- Lacks diversity on complex attack parameters

The main weakness of this paper is the existence of many prior works on backdoor frameworks, specifically TrojanZoo. The difference between the proposed system and TrojanZoo appears to be minor. More comments on this "Relation to Prior Work."

Another concern is that the BackdoorBench only offers parameter change on several attack parameters (poison ratio). However, backdoor attacks have many additional parameters depending on the attack specifics, e.g. trigger size, physical triggers, perturbation size. There are other attack setups such as label-specific backdoors, where attacker only backdoors data from one label. I would appreciate if the authors can add these parameters in the benchmark, so that the defenses do not only optimize for a biased set of attack parameters.


**Additional Feedback:**

None.

**Clarity:**

The paper has some language issue making it hard to understand. I recommend that the authors proofread the paper multiple times to fix language/presentation issues.


**Correctness:**

The implementation is correct and designs are appropriate.


**Documentation:**

The code is well documented with sufficient information to reproduce or add defenses/attacks.

**Ethics:**

No.

**Relation To Prior Work:**

TrojanZoo is a popular framework used in prior work, and many research groups are already familiar with the setup. It is unclear to me whether this paper adds additional benefits over TrojanZoo, as the only difference focuses on the number of defenses/attacks. While I appreciate the authors running 8,000 pairs of evaluations, it is unclear whether those tests produce significant findings that we did not already know. Finally, given how people are familiar with TrojanZoo codebase already, the benefit for researchers to switch to BackdoorBench appears to be low.


**Summary And Contributions:**

The paper presents a benchmark and software system to evaluate poison attacks and defenses. The authors include a large number of popular defenses and attacks. The authors further make it easy to implement additional attacks and defenses.

---

> ### Author Response · Authors · 2022-08-12
> **Response to Reviewer RPfC: greatly appreciate your constructive comments and high recognition about our efforts (Part 3)**
>
>
> **Q3**: The paper has some language issue making it hard to understand. I recommend that the authors proofread the paper multiple times to fix language/presentation issues.
>
> **R3:** Greatly appreciate your careful inspection. We will seriously follow your suggestion to thoroughly proofread the manuscript to fix language/presentation issues.
>
> ---
>
>
> To avoid too long wait, we firstly post the above preliminary responses to your concerns. All additional results/analysis promised above will be immediately added here and in the revised manuscript, once they are finished. Hope these responses helpful to address your concerns. And, we are willing to discuss with you about any further concerns.
>
>
>
> Thanks again for your constructive comments and your recognition of our efforts.
>
> Sincerely,
>
> Authors

---

> ### Author Response · Authors · 2022-08-12
> **Response to Reviewer RPfC: greatly appreciate your constructive comments and high recognition about our efforts (Part 2)**
>
> **Q2: Another concern is that the BackdoorBench only offers parameter change on several attack parameters (poisoning ratio)**. However, backdoor attacks have many additional parameters depending on the attack specifics, e.g. trigger size, physical triggers, perturbation size. There are other attack setups such as label-specific backdoors, whereattacker only backdoors data from one label. I would appreciate if the authors can add these parameters in the benchmark, so that the defenses do not only optimize for a biased set of attack parameters.
>
> **R2:** Thanks for this constructive suggestion.
> To answer the concern about the hyper-parameter tuning from Reviewer wFZF, we have presented a detailed explanation above (please refer to the Response R1 to Reviewer wFZF). We think that concern is related to your concern. For clarity, here we borrow some key points from the Response R1 to Reviewer wFZF and reorganize the explanations as follows:
>
>
> - We didn't search a optimal value or try different values of each hyper-parameter in our evaluations, mainly due to two reasons.
> 	+ If we do that, then there will be an explosive growth of the number of evaluations. We think you may understand that, as a young research team at univerisity, we don't have enough computing resources to finish too many evaluations in limited time. However, we realize that this should not an excuse, since we aim to build a solid benchmark with comprehensive evaluations. We will try our best to invest more resources to add more evaluations continuously.
> 	+ Meanwhile, we think that it is unfair and makes no sense in practice to search the optimal value of each hyper-parameter in the evaluations. Because there is no objective rule to determine a good value of each hyper-parameter in practice. We should not only focus on the best ACC/ASR of one attack/defense method, but also be care about its sensitivity to some key hyper-parameters, which could reflect its practical usage.
>
> - **How do we set the hyper-parameter values in our current 8000 pairs of evaluations.** We would like to invite you to see the detailed explanation in the Response R1 to Reviewer wFZF, and here we don't repeat. The only emphasize is that the analysis based on our current 8,000 evaluations will not be largely affected by the variations of other hyper-parameters.
>
> - **Why do we choose the poisoning ratio as the major varying factor in our evaluations?**
>  The poisoning ratio is a common factor that is aganostic to the attack method, the dataset, the model architecture, and we found that lots of backdoor attack works tried different poisoning ratios in their experiments. Thus, the analysis *w.r.t.* the poisoning ratio is applied to all methods, and can reveal some general findings. In contrast, some hyper-parameters are particular for only one or a few methods. For example, the trigger size only exists in the backdoor attack with local triggers, such as BadNets. Besides, since TrojanZoo have presented several good analysis about the trigger, we try to provide some new analysis from the general perspectives that are suitable for all methods. As demonstrated in the last part of the common response, we are conducting several analysis from other general perspectives, and partial results will be posted in the response window before the rebuttal deadline.
>
> - **Our plan for evaluations with changes on more hyper-parameters.** As you mentioned, the evaluations with changes on several hyper-parameters are also important to understand the real performance of one method. Thus, for each attack/defense method, we will pick the key hyper-parameters (*e.g.*, the trigger size in BadNets, the alpha blending coefficient in Blended), and test the method's sensitivity to these hyper-parameters, as well as the transferability of good hyper-parameter values across different datasets, different model architectures, as well as different opponents (*i.e.*, defense/attack methods). These tests will be valuable to understand the real performance of one attack/defense method in practice. We will present partial evaluations and analysis before the rebuttal deadline, and plan to present  all sensitivity evaluations and analysis in the website of BackdoorBench when they are finished.
>
> In terms of the comment "**There are other attack setups such as label-specific backdoors, where attacker only backdoors data from one label**", we are not sure that we have understand it correctly. If it means the setting of clean label, where only the clean samples from the target class are poisoned, then actually we have included the method with this setting, *i.e.*, the LC attack (see Table 1 in the main manuscript). If our understanding is incorrect, your further clarification will be greatly appreciated, and we are willing to add any suggested evaluations.

---

> ### Author Response · Authors · 2022-08-12
> **Response to Reviewer RPfC: greatly appreciate your constructive comments and high recognition about our efforts (Part 1)**
>
> Dear Reviewer RPfC,
>
> We sincerely appreciate your precious time and constructive comments, and are greatly encouraged by your high recognition about our efforts to build a new backdoor benchmark.
> In the following, we would like to answer your concerns separately.
>
> ---
>
> **Q1: Relation To Prior Work**: TrojanZoo is a popular framework used in prior work, and many research groups are already familiar with the setup. It is unclear to me whether this paper adds additional benefits over TrojanZoo, as the only difference focuses on thenumber of defenses/attacks. While I appreciate the authors running 8,000 pairs of evaluations, it is unclear whetherthose tests produce significant findings that we did not already know. Finally, given how people are familiar with TrojanZoo codebase already, the benefit for researchers to switch to BackdoorBench appears to be low.
>
> **R1:** Thanks for this insightful comment. We sincerely admire the efforts and contributions of all pioneer backdoor benchmarks, especially the great benchmark TrojanZoo. Actually we have learned a lot from them, and we always think about what new valuable contributions we can make to the backdoor community.
>
> We would like to invite you to see **the common response** named "**A common response to the concern about the diff erencebetween BackdoorBench and TrojanZoo**" posted at the top location for more details, where we demonstrate the significant diferences in codebase, analysis and findings between our BackdoorBench and TrojanZoo, as well as our motivation, the latest progress and our future schedule. Hope those responses be helpful to evaluate the contribution of our benchmark to the backdoor community.

---

> ### Author Response · Authors · 2022-08-19
> **Latest response to Reviewer RPfC: Thanks for your help, and all suggested updates have been added**
>
> Dear Reviewer RPfC,
>
> We sincerely appreciate your constructive comments and warming encouragements. All suggested updates by you and other reviewers have been added into the revised manuscript and the codebase of BackdoorBench, and are summarized in the latest common response posted above.
>
> Hope our previous responses and all updates helpful for you to measure the value of our work. We are looking forward to your further feedback, and are willing to answer any further concern. Thanks.
>
> Best regards,
>
> Authors

---

### Official Review · Reviewer_wFZF · 2022-07-27
**Solid benchmarking paper with promising code base**

**Rating:** 9
**Confidence:** 3
**Correctness:** It is not always clear how hyperparam…

**Strengths:**

The work is an excellent contribution toward trustworthy machine learning for the underrepresented backdoor attack vector. It provides a reasonable set of attacks and defenses and gives a clear presentation of the categorization of these attacks and defenses. In addition, it goes further by providing a set of explainability methods.

**Weaknesses:**

During the evaluation it is not always clear how the hyperparameters are chosen. For example, whether a separate hyperparameter search is performed for each attack and defense to find the optimal case.
In section 4.3, I am not sure that the explanation for decreasing poisoning success with increasing number of poisons is sufficient. It could also be a matter of different training settings/conditions.

**Additional Feedback:**

More trials per run in Figure 3 would make potential trends clearer.

**Clarity:**

The paper has some spelling and grammatical mistakes that should be revised, but overall is clear. For example:

- However, we find that the evaluations of new methods are often insufficient, with comparisons with limited previous methods, based on limited models and datasets.
- stat-of-the-art
- evluate

**Documentation:**

The code has its own website, which provides a comprehensive overview and explanations of backdoor attacks and defenses. The linked code base contains a detailed description of how to use the code and descriptions of the implemented methods.

**Ethics:**

I could not identify any ethical concerns.

**Relation To Prior Work:**

Prior related work is sufficiently described.

**Summary And Contributions:**

The paper presents a backdoor benchmarking framework. It includes several basic and current attacks and defenses that target a model's integrity during the training phase of a model. The paper includes evaluating these methods and further analyzing the effect of different poison ratios and the effect of attacks and defenses. To facilitate interpretation of benchmark results, several explanatory methods are included in the framework.

---

> ### Author Response · Authors · 2022-08-11
> **Response to Reviewer wFZF: greatly appreciate your constructive comments and high recognition about our efforts (Part 2)**
>
> **Q2: In section 4.3, I am not sure that the explanation for decreasing poisoning success with increasing number of poisons is sufficient. It could also be a matter of different training settings/conditions.**
>
> **R2:** Thanks for this insightful comment. We would like to share our thoughts from the following aspects:
> - When we study the effect of poisoning ratio in Section 4.3, we fix all other conditions. And, the detailed analysis presented in Section 4.3 and Figure 4 has revealed the behind reason. Thus, the conclusion that the backdoor attack with higher poisoning ratio may lead to lower attack success rate under some backdoor defenses is reliable.
> - However, we guess your meaning may be that not only the poisoning ratio, some other factors could also lead to the similar phenomenon. We fully agree with this point. We think that the finding *w.r.t.* the poisoning ratio could be extended to a more general assertion: ***the stronger backdoor attack in the case of no defense may achieve lower attack success rate under some backdoor defenses***. We believe it is reasonable, as the stronger attack means the stronger connection between the trigger and the target class, implying the larger difference between poisoned and clean samples in the backdoored model. As many defense methods just adopt some differences between poisoned and clean samples (*e.g.*, different activation paths adopted in FP, different training loss values adopted in ABL), the larger difference means that the stronger attack could be more easily detected or removed by these defenses. There should be other factors which could lead to stronger attack in the case of no defense, *e.g.*, manipulating the feeding frequency or order of poisoned samples during the training process, or choosing some samples to poison via some rules rather than randomly. We will explore and verify this assertion in the future, and we think it could inspire more researchers to think about what is a really strong attack/defense.
>
> ---
>
> **Q3: More trials per run in Figure 3 would make potential trends clearer.**
>
> **R3:** Thanks for this constructive suggestion. We are running this suggested experiments, and will update Figure 3 in the revised manuscript when these experiments are finished (maybe within 5 days, *i.e.*, before August 16, 2022). We will keep you informed about the progress.
>
> ---
>
> **Q4: The paper has some spelling and grammatical mistakes that should be revised, but overall is clear.**
>
> **R4:** Greatly appreciate your careful inspection, and we will thoroughly proofread the manuscript to correct spelling and grammatical mistakes and update in the revised manuscript.
>
> ---
>
> To avoid too long wait, we firstly post the above preliminary responses to your concerns. All additional results/analysis promised above should be immediately added here and in the revised manuscript, once they are finished. Hope these responses helpful to address your concerns. And, we are willing to discuss with you about any further concerns.
>
> Thanks again for your recognition of our efforts and constructive comments.
>
> Sincerely,
>
> Authors

---

> > ### Comment · Reviewer_wFZF · 2022-08-15
> > **Response**
> >
> > Thank you for the detailed comments and explanations. I have read the response and it helps to clarify open questions. I agree with the additional explanation about the drop in performance when more poisons are used. The reasoning that more poisons leads to a greater difference between poisoned and clean samples is now more intuitive.
> >
> > I also agree that it is impossible to test all possible hyperparameters. I pointed this out to make sure the parameter choices were transparent, and I must admit that I initially overlooked the hyperparameters in the supplemental material.

---

> > > ### Author Response · Authors · 2022-08-19
> > > **Thanks for your help, and all suggested updates have been added**
> > >
> > > Dear Reviewer wFZF,
> > >
> > > We sincerely appreciate your constructive comments and warming encouragements. All suggested updates by you and other reviewers have been added into the revised manuscript and the codebase of BackdoorBench, and are summarized in the latest common response posted above.
> > >
> > > Best regards,
> > >
> > > Authors

---

> > > > ### Comment · Reviewer_wFZF · 2022-08-23
> > > > **Response**
> > > >
> > > > Thank you for the update. I checked the updated paper and have no further comments.

---

> > > > > ### Author Response · Authors · 2022-08-24
> > > > > **Thanks for your help**
> > > > >
> > > > > Dear Reviewer wFZF, thanks for your help.

---

> ### Author Response · Authors · 2022-08-11
> **Response to Reviewer wFZF: greatly appreciate your constructive comments and high recognition about our efforts (Part 1)**
>
> Dear Reviewer wFZF,
>
> We sincerely appreciate your precious time and constructive comments, and are greatly encouraged by your high recognition about our efforts to build a new backdoor benchmark.
> In the following, we would like to answer your concerns separately.
>
> ---
>
> **Q1: During the evaluation it is not always clear how the hyper-parameters are chosen. For example, whether a separate hyper-parameter search is performed for each attack and defense to find the optimal case.**
>
> **R1:** Thanks for this constructive comment. We would like to explain it from the following three aspects.
>
> - **Actually we didn't perform a separate hyper-parameter search for each method**.
> 	+ As shown in Tables 1 and 2 in Appendix, most methods have several hyper-parameters. For most hyper-parameters of a method, there is neither a good rule to determine the values, nor a suitable range of the values suggested in its original manuscript. And, the suitable value or range of each hyper-parameter may vary across different datasets, different model architectures, different against attack/defense methods. Consequently, the hyper-parameter search space for each method could be very large, requiring unimaginably high computational resourse.
> 	+ Even assuming sufficient computing resources, then we can search a good value for each hyper-parameter of each method in each evaluation. However, the comparison results and analysis based on sufficient hyper-parameter search may be unfair and make no sense in practice. Because, we still cannot tell a rule or even some experiences to determine the hyper-parameter values in practice. The sensitivity to hyper-parameters should also be an important metric of one method's performance, not just the best ACC/ASR values through the sufficient hyper-parameter search.
>
> - **How do we set the hyper-parameter values in our current 8000 pairs of evaluations**.
> 	+ Firstly, all hyper-parameter values of each method adopted in our evaluations have been clearly written in Tables 1 and 2 in Appendix. All reported results can be easily re-produced.
> 	+ If the original paper has provided the suggested good values of some hyper-parameters, then we adopt those values in our evaluations. For example, the ANP defense method explicitly wrote that "*the perturbation budget  $\epsilon = 0.4$ and the trade-off coefficient $\alpha = 0.2$*", so we also adopt these values in our evaluations.
> 	+ For those hyper-parameters without suggested values/ranges (or even without descriptions) in their original papers, we will search values that lead to comparable results (ACC/ASR) with the reported results in the same setting (*i.e.*, same dataset, same/similar model architecture, same poisoning ratio), then fix these values in evaluations of other settings (*e.g.*, changing the poisoning ratio).
> 	+ The consistent values of hyper-parameters of each method across different settings somewhat guarantee the fairness of evaluations. And, since the adopted values may not be the optimal ones for some hyper-parameters, we didn't conduct the fine-grained analysis about the effects of some specific hyper-parameters (*e.g.*, the trigger size/location in attack methods with patch based triggers). Instead, we provided some high-level analysis *w.r.t.* the shared hyper-parameters in all methods (*e.g.*, the number of classes, the poisoning ratio, the model architecture). The findings of these high-level analysis will not be significantly affected by the particular hyper-parameters of each individual method.
>
> - **Sensitivity test *w.r.t.* key hyper-parameters of each method**. As claimed above, the sensitivity to hyper-parameters should also be an important metric of one method's performance. Thus, as described in our future schedule (see the last part of the common response to all reviewers), we plan to conduct the sensitivity test for each attack/defense method.  **Before the end of the rebuttal period, we will post partial sensitivity results here**, and the complete sensitivity tests of all methods will be posted in the website of BackdoorBench (*i.e.*, https://backdoorbench.github.io/). The comprehensive sensitivity test will help the community to better understand and measure the performance of each method.

---

### Author Response · Authors · 2022-08-10
**A common response to the concern about the difference between BackdoorBench and TrojanZoo (Part 4)**

### **4\. The latest progress and future schedule of BackdoorBench**

Following the above rules, we are updating BackdoorBench from several aspects (see 4.1 below), and have made the future schedule (see 4.2 below).


####  **4.1 Our latest progress during the rebuttal stage**
- **Codebase**:
   + **New methods**: we are adding the implementations of 2 backdoor defense methods (MCR and I-BAU) and 1 backdoor attack method (LIRA).
   + **New domain**: we are adding the implementations of several bakcdoor learning methods for the natural language processing (NLP) domain.
   + **New data loader**: we are adding the parallel data loader to support larger scale experiments (*e.g.*, the full ImageNet).
   + **New attack setting**: we are adding the all2all attack setting, *i.e.*, there is one trigger for each class.
- **Evaluation**:
   + **New dataset**: we are running the evaluations on the full ImageNet.
   + **New attack setting**: we are running the evaluations of the all2all attack setting.
   + **New model architectures**: we are running the evaluations on the ViT model.
   + **New applications (NLP)**: we are running the evaluations on the NLP tasks.
   + **Statistical test of Fig. 3**: we are repeating the evaluations shown in Fig. 3 with several random initializations, and will update Fig. 3 with error bars in the revised manuscript.
- **Analysis**:
	+ **Quick learning of backdoor**: we find a general phenomenon that the backdoor could be quickly learned within a few epochs. We are presenting a demo of analyzingthis phenomenon.
	+ **Trigger generalization**: in current setting, a default setting is that the trigger used in the testing and training stages is same. However, we find that there exsit some testing triggers different with the training trigger could also successfully activate the backdoor. We call this phenomenon as *trigger generalization*. We are presenting a demo of analyzing this phenomenon.
	+ **Frequency saliency map**: we propose a novel visualization tool, called *frequency saliency map* (FSM), from the frequency space. FSM visualizes the salient frequencies of one image that contribute most to the prediction, and provides a novel frequency perspective to review backdoor learning. The detailed definition of FSM will be added into the revised manuscript, and we are presenting a demo to show the new insight of backdoor learning using FSM.
	+ **Memorization and forgetting of poisoned samples during the training**: since it seems that the backdoored model has memorized the trigger pattern, we are preparing a demo of analyzing the memorization and forgetting of poisoned samples during the training process, utilizing the membership inference attack technique.

**Note**: The details of each above term will be continuously added here or at the separate responses to some reviewers, or in the revised manuscript, when it is fully finished during the rebuttal period.

#### **4.2 Our future schedule**
- **Codebase**:
	+ **New methods**: we are always keeping track of all new backdoor learning methods that are published at top-tier journals/conferences in the AI and Security community, and will add their implementations into BackdoorBench.
	+ **New domains**: we plan to expand BackdoorBench from the current vision domain to other domains, such as NLP, Speech and Reinforcement learning.
	+ **Pip install**: we plan to make the codebase of BackdoorBench as a Python package, such that it can be easily installed via pip. It will facilitate more researchers to use it.
- **Evaluation**:
	+ **Evaluations on new methods, new datasets and new model architectures**: we plan to add more widely used datasets and more mainstreamed model architectures to evaluate all existing and new implemented backdoor learning methods.
- **Analysis**:
	+ **Sensitivity to key hyper-parameters of each method**: each attack/defense method has some hyper-parameters, and their performance will be affected by these hyper-parameters. We will analyze the sensitity of each attack/defense method to its key hyper-parameters, and the variation of good hyper-parameter values across model architectures, datasets.
	+ **Poisoning sample’s effect**: we will analysze the effect of different poisoned samples, given the same poisoning ratio.
	+ **Assumption grouping of different attack/defense methods**: according to our evaluations, we found that some methods always have similar performance (such as FT/FP/NAD/NC, see Sec. 4.3). It implies that there should be some similarity/redundancy among their assumptions. We will explore such similarity and group similar methods together.
	+ **Strongly correlated pairs of attack and defense assumptions**: we found that some attack methods can always evade some defense methods, or can always be defended by some defense methods. It implies that their assumptions have strong correlations. We will explore such strongly correlated pairs of assumptions.

Sincerely,

Authors

---

### Author Response · Authors · 2022-08-10
**A common response to the concern about the difference between BackdoorBench and TrojanZoo (Part 3)**

####  **2.2 Differences in analysis and findings**
   + **TrojanZoo** has provided very abundant and diverse analysis of backdoor learning, mainly including:
		+ Attack: the effects of trigger size, trigger transparency, data complexity, backdoor transferability to downtream tasks
		+ Defense: the tradeoff between robustness and utinity, the tradeoff between detection accuracy and recovery capability, the impact of trigger definition
   + **BackdoorBench** have provided and are preparing the following analysis:
		+ **Section 4.3, Effect of poisoning ratio**: we found that the ASR of backdoor attacks under defense doesn't consistently increase along with the increase of poisoning ratio, and there is a sharp decrease when the poisoning ratio exceeds some threshold. Our deep analysis reveals that in the case of large poisoning ratio, the difference between poisoned and clean samples will be significant to be easily defended. It inspires to design more advanced backdoor attacks with smaller poisoning ratio.
		+ **Section 4.2 and Fig. 2, Effect of number of classes**: we found that for a dataset with larger number of classes, it is more difficult to recover the correct prediction after backdoor defense, which is measured by the robust accuracy (R-Acc). It inspires to think about the better defense for large-scale dataset like the full ImageNet.
		+ We are analyzing **the quick learning of backdoor**, which could reveal the reason of backdoor formution and be utilized to develop more advanced backdoor attack and defense methods.
		+ We are analyzing the **trigger generalization**, which could demonstrate the behind relationship between the triggers and the target class, and it may affect the usage of backdoor learning in the application of intellectual property protection.
		+ We are analyzing the **memorization and forgetting of poisoned samples** during the training, which could reveal the process of backdoor formation, and could be used to develop more durable backdoors or removing the backdoor through forggeting.
		+ We have provided **several analysis tools**, including: GradCAM, Shapley value, t-SNE, Neuron activation, frequency saliency map, gradient trajectory, *etc.*

We think the analysis and findings provided by both TrojanZoo and our BackdoorBench will be beneficial to the backdoor learning field.

---

### **3\. What characteristics should a desired backdoor benchmark have?**

We think a desired backdoor benchmark should have the following characteristics:

* **Codebase:** The codebase should be user-friendly (easy to learn for beginners), easily extensible to implement new settings/algorithms/datasets/model-architectures, and lightweight (avoiding redundant codes).
* **Evaluation:** The benchmark should provide comprehensive and fair evaluations to facilitate the comparisons with existing methods and evaluating the performance of new methods.
* **Analysis:** The benchmark should provide deep analysis from different perspectives, and reveal new insights to inspire more researchl; Rich analysis tools should be also provided to facilitate other researchers.
* **Maintenance and update:** The benchmark should be actively maintained (_e.g._, answering and solving the issues posed by readers), and regularly updated (self updating and merging the third-parties' pushes).

---

### Author Response · Authors · 2022-08-10
**A common response to the concern about the difference between BackdoorBench and TrojanZoo (Part 2)**

### **2\. What are the differences between our benchmark and TrojanZoo? / What unique values and contributions of our benchmark?**

We sincerely admire the contributions of all pioneer backdoor benchmarks, especially TrojanZoo. Actually, we have learned a lot from it when we build BackdoorBench. As TrojanZoo has provided very good implementations and presented several insightful analysis, it motivates us to think about what new values we can provide in the new benchmark. In the following, we would like to objectively summarize the main differences on the codebase and analysis between TrojanZoo and BackdoorBench, without any subjective judgement.



####  **2.1 Differences in codebases**
   - **Programming Style**: Object-Oriented Programming (OOP) *vs.* Procedural Oriented Programming (POP). Both benchmarks adopt the modular design to ensure easy extensibility. However, TrojanZoo adopts the OOP style, and it defines each module as one class (see the folder *./trojanzoo/* of the TrojanZoo Github repository), and when implementing a specific algorithm, there will be several hierarchical instances and inheritance of classes. In contrast, our BackdoorBench adopts the POP style, where each module is defined as one function, and each specific algorithm is implemented by several functions in a streamline. Take the attack algorithm *BadNets* as an example,
     + In **TrojanZoo**, it is started from the file '*./trojanvision/attacks/backdoor/normal/badnet.py*', where there is only one class name, and it is a child class of the class *BackdoorAttack* defined in '*./trojanvision/attacks/abstract.py*'. And, the class *BackdoorAttack* also inherits something from the class *Attack*, which is defined in '*./trojanzoo/attacks.py*'.
     + In **BackdoorBench**, the BadNets method is fully implemented in the file '*./attack/badnet_attack.py*'. There are two blocks of the code, including the data preparation block and the training block. More fine-grainedly, there are 7 functions in a streamline, including: (1) config args, save_path, fix random seed; (2) set the clean train data and clean test data; (3) set the attack image transform and label transform; (4) set the backdoor attack data and backdoor test data; (5) set the device, model, criterion, optimizer, training; (6) attack or use the model to fine-tune with 5% clean data; (7) save the attack result for defense. The prototyps of most functions are defined in the shared fold '*./utils*', with different parameters for each specific algorithm.
   - **Learning cost for beginners**: when one beginner starts to learn and use one algorithm in a benchmark: in BackdoorBench, he/she just needs to read the code in one file, and he/she can know the complete procedure of this algorithm, and the procedures of other algorithms are similar; in TrojanZoo, he/she will jump hierarchically to several files to learn the definitions of hierarchical classes and their inheritance relationships. We can claim that the learning cost of BackdoorBench for beginners is lower than that of TrojanZoo.
   - **Extensibility**:
     + Both OOP and POP have their own advantages. The OOP style is structured and facilitate the operation and maintenance of experienced engineers. It is suitable for a project with clearly defined settings and functional requirements. The POP style is clear and flexible, and suitable for a project with new diverse settings and functional requirements. Specifically for backdoor learning, it is still at the rapid development stage, and lots of new settings are constantly emerging. In this case, we think the POP style is a more suitable choice, and the new setting/algorithm/dataset/model could be quickly implemented by adjusting the different functions based on the existing implementations.
     + Besides, due to the different learning costs for beginners (see the above point), a user can easily implement his/her own algorithm after quick learning one existing algorithm in BackdoorBench, while he/she has to be familiar with the whole framework of TrojanZoo before implementing his/her own algorithm based on TrojanZoo.

---

### Author Response · Authors · 2022-08-10
**A common response to the concern about the difference between BackdoorBench and TrojanZoo (Part 1)**

Dear Reviewers and AC,

We are deeply encouraged by the positive comments from all reviewers about our efforts in building a comprehensive benchmark for backdoor learning. Since we are not the first attempt in the backdoor community, it is very important to clarify the differences between our benchmark and existing benchmarks (especially TrojanZoo), such that the unique value and contribution to the community could be well evaluated.  We fully agree with the reviewers that the date and number of implemented methods should not the most important and distinguishable value of a new benchmark, as they can be easily updated by later benchmarks.

We sincerely appreciate this precious opportunity to clearly demonstrate our thoughts about building the new benchmark. We would like to expand from the following four aspects:


1. **Why do we need to build a new benchmark?**
We will explain it from our initial motivation to a broader view to evaluate the value of building a new benchmark.
2. **What are the differences between our benchmark and TrojaZoo?** In other words, **what unique values and contributions** can we provide to the backdoor learning?
We will present their differences on codebase and the provided analysis.
3. **What characteristics should a desired backdoor benchmark have?**
We will answer this question from four aspects, including codebase, evaluations, analysis, as well as maintenance and update.
4. **The latest progress and future schedule of BackdoorBench**.

---


### **1\.Why do we need to build a new benchmark?**

Considering that there have been one or a few benchmarks in the community, is it still valuable to build new benchmarks? We would like to explain it from the following two aspects.

* **Our initial motivation**: when we started the research on backdoor learning from 2 years ago, we have witnessed that more and more new backdoor learning methods are being quickly proposed, as well as their diverse settings (_e.g._, different threat models, different types of triggers, different datasets, different model architectures). **(1)** Due to the quick development, many new methods have not been implemented in a unified framework, causing the difficulty to fairly compare with existing methods for new research. Besides, due to the diverse settings, there are lack of consistent and comprehensive evaluations of existing attacks against existing defenses. Consequently, **it is difficult to track the real progress of this field, and to correctly evaluate the new methods**. **(2)** Moreover, several new interesting phenomenons about backdoor learning have been observed in other works or our own research, such as the quick learning of backdoor or the generalization of triggers (see the details in later responses). **The verification and the reason exploration of these interesting phenomenons should be built upon the comprehensive evaluations of existing works**. Above two points motivate us to build a new backdoor benchmark.
* **A broader view to evaluate the value of building a new benchmark**. **(1)** Recall the development history of other fields, we can see that the existence of multiple benchmarks/toolboxes is a common phenomenon, and is beneficial for the whole community, especially at the early development stage of one field. For example, in the deep learning framework, there have been Caffe, Tensorflow, Pytorch, PaddlePaddle, _etc._; for adversarial examples, there are Cleverhans, Foolbox, RobustBench, RobustART, _etc._ The moderate competition among several benchmarks could **not only promote the development speed and maintenance frequency** of each benchmark, **but also encourage each benchmark to provide high-quality, distinct and diverse services**. **(2)** In terms of backdoor learning, it is still at the early stage and has attracted wide attention. We think that the backdoor community will be happy to see more competitors. And, **we are confident that our new benchmark is a strong competitor to make valuable contributions to this field**, as it provides several new implementations, new analysis and new analysis tools, to facilitate and inspire new research (please refer to the details in the answer to the second question).

---

### Author Response · Authors · 2022-08-19
**A latest common response: summary of all updates**

Dear Reviewers and AC,

According to all reviewers' constructive suggestions, we have made substantial updates in both the manuscript and the codebase of BackdoorBench.
For convenience, we merge the main manuscript and appendix into one revised PDF file, and highlight all updates in blue. We summarize the major updates as follows.

**Major updates in the revised manuscript**:
- Differences between our BackdoorBench and TrojanZoo: Line 115-134 of Section 2;
- Refinements of Tables 2 and 3 with more clear definitions and notations: Page 4;
- Moving limitations and societal impacts into the main manuscript: Line 292-299 of Section 5;
- Illustrations of hyper-parameter's settings: Line 602-637 of Section A.3;
- Detailed differences between our BackdoorBench and TrojanZoo: Section B;
- Effect of poisoning ratio based on evaluations with multiple random trials: Section C.5;
- Sensitivity to hyper-parameters: Section C.6;
- Analysis of quick learning of backdoor: Section C.7;
- Analysis of memorization and forgetting of backdoors: Section C.8;
- Analysis of trigger generalization of backdoor attacks: Section C.9;
- Evaluation on vision transformer: Section C.10;
- Evaluation on ImageNet: Section C.11;
- Visualization tools and results: Section C.12;
- BackdoorBench in Natural Language Processing: Section D.

**Major updates in the codebase (https://github.com/SCLBD/BackdoorBench)**:
- Implementing the evaluation on ImageNet: all files under the folder `for_imagenet/`;
- Implementing the evaluation on ViT: all files under the folder `attack/`;
- Simplifying the dataloader codes: all files under the folder `attack/`;
- Implementations of backdoor learning in the NLP domain: all files under the folder `backdoorbench_nlp/`;
- Adding one visualization tool *frequency saliency map* and presenting a demo: `visualization/visualize_fre.py`, `visualization/Frequency_demo.ipynb`;
- Implementing all2all backdoor attack: `utils/aggregate_block/bd_attack_generate.py`, `utils/backdoor_generate_pindex.py`;
- Implementing new methods: the LIRA attack `attack/lira_attack.py`, the I-BAU defense `defense/i-bau/`.

Finally, we would like to sincerely appreciate all reviewers for their precious time, insightful and constructive suggestions, and warming encouragements, which have significantly improved the quality of our benchmark, compared to the initial submitted version. We will make continuous efforts to maintain and expand BackdoorBench to facilitate and inspire more future researches in backdoor learning.

Best regards,

Authors

---

### Meta-Review · Area_Chair_RS9Y · 2022-09-14

**Recommendation:** Accept
**Confidence:** 4

**Metareview:**

This is a valuable benchmark on backdoors. Most reviewers argue for acceptance, some strongly so (with scores 7,7,8,9), while only one reviewer gave a rejecting score. That reviewer mostly questioned the relationship to TrojanZoo, and the authors discussed this comprehensively now. That reviewer, RPfC, also was inactive during the rebuttal and decision process, and thus I do not weigh their (apparently answered) concerns highly. I thus recommend acceptance.

---

### Decision · Program_Chairs · 2022-09-16

Accept